# Electrical spectroscopy of polaritonic nanoresonators

Sebastián Castilla [1,12] ✉, Hitesh Agarwal[1,12], Ioannis Vangelidis[2,12], Yuliy V. Bludov [3,4,12], David Alcaraz Iranzo [1], Adrià Grabulosa[1], Matteo Ceccanti[1], Mikhail I. Vasilevskiy [3,4,5], Roshan Krishna Kumar [1], Eli Janzen[6], James H. Edgar [6], Kenji Watanabe [7], Takashi Taniguchi [8], Nuno M. R. Peres [3,4,5,9], Elefterios Lidorikis [2,10] & Frank H. L. Koppens [1,11] ✉

One of the most captivating properties of polaritons is their capacity to confine light at the nanoscale. This confinement is even more extreme in two-dimensional (2D) materials. 2D polaritons have been investigated by optical measurements using an external photodetector. However, their effective spectrally resolved electrical detection via far-field excitation remains unexplored. This hinders their exploitation in crucial applications such as sensing, hyperspectral imaging, and optical spectrometry, banking on their potential for integration with silicon technologies. Herein, we present the electrical spectroscopy of polaritonic nanoresonators based on a high-quality 2D-material heterostructure, which serves at the same time as the photodetector and the polaritonic platform. Subsequently, we electrically detect these mid-infrared resonators by near-field coupling to a graphene pn-junction. The nanoresonators simultaneously exhibit extreme lateral confinement and high-quality factors. This work opens a venue for investigating this tunable and complex hybrid system and its use in compact sensing and imaging platforms.

Polaritons are coupled excitations of electromagnetic waves with charged particles (plasmons polaritons)[1,2] or lattice vibrations (phonon polaritons)[3–5]. The polaritonic properties become extreme in two-dimensional (2D) materials, including wavelength confinement by factors up to 300[6,7], ray-like propagating modes[8–10], long lifetimes[11,12], and capabilities to tune its properties in situ[4,13]. These polaritons have been investigated in near-field studies (e.g., using scanning near-field optical microscopy)[8,9,14–21], by electron energy loss spectroscopy (EELS)[22–24] and far-field with Fourier Transform Infrared spectroscopy (FTIR)[1,6,25–34], which, however, constitute bulky systems that require a typical cooled external detector. In order to achieve a highly compact platform, 2D polaritons have been electrically detected by using a graphene nanodisk array[35] or antennas that launch hyperbolic phonon polaritons (HPPs) of hBN in the detector's photoactive area[36]. However, the detection is mainly based on increasing the magnitude of the photoinduced signal at a fixed incident wavelength[35,37–39], at the expense of spectral information.

Electrical spectroscopy of polaritonic nanoresonators is a unique capability with prospects for nano-optoelectronic circuits, and molecular sensing applications[33,40], as they can be strongly coupled to

[1]ICFO - Institut de Ciències Fotòniques, The Barcelona Institute of Science and Technology, Castelldefels (Barcelona), Spain. [2]Department of Materials Science and Engineering, University of Ioannina, Ioannina, Greece. [3]Centro de Física (CF-UM-UP), Universidade do Minho, Braga, Portugal. [4]Departamento de Física, Universidade do Minho, Braga, Portugal. [5]International Iberian Nanotechnology Laboratory (INL), Braga, Portugal. [6]Tim Taylor Department of Chemical Engineering, Kansas State University, Manhattan, KS, USA. [7]Research Center for Electronic and Optical Materials, National Institute for Materials Science, Tsukuba, Japan. [8]Research Center for Materials Nanoarchitectonics, National Institute for Materials Science, Tsukuba, Japan. [9]POLIMA-Center for Polariton-driven Light-Matter Interactions, University of Southern Denmark, Odense M, Denmark. [10]University Research Center of Ioannina (URCI), Institute of Materials Science and Computing, Ioannina, Greece. [11]ICREA - Institució Catalana de Recerca i Estudis Avançats, Barcelona, Spain. [12]These authors contributed equally: Sebastián Castilla, Hitesh Agarwal, Ioannis Vangelidis, Yuliy V. Bludov. ✉e-mail: sebastian.castilla@icfo.eu; frank.koppens@icfo.eu

molecular vibrations[27,28,41]. Here, we merge 2D polaritonic resonators with a graphene pn-junction into a single high-quality 2D-material heterostructure to realize the electrical spectroscopy of deep sub-wavelength polaritonic nanoresonators. The quality factor of the nanoresonators and the high mobility of graphene play key roles, as high values enable effective photodetection of polaritonic resonances, in contrast to low-quality factor values in low mobility[6,30–32] or patterned graphene[1,25–27,33,42], resulting in a diminished detection efficiency and spectral resolution. Our approach eliminates the need for an external detector for spectroscopy and leads to device miniaturization. Its small photoactive area (comparable to the hot carriers cooling length of ~ 0.5 to 1 μm[36,43]) is adequate in converting the incoming light into an electrical signal[36], contrary to FTIR, which requires large optically active device areas ($\gtrsim 30 \times 30$ μm²) to obtain a reasonable signal-to-noise ratio[6,32].

Our methodology has enabled us to investigate the contribution of the hybridized modes present at the underexplored hBN lower reststrahlen band (RB) in the photocurrent spectrum, which corresponds to a different type of hyperbolicity (type I) with respect to the typically studied (type II) for the upper RB[3,4]. In fact, at the lower RB spectral range, we have observed the highest Q-factors and lateral confinement among the whole investigated mid- and long-wave infrared spectra. The active tunability of the polaritonic nanoresonators' spectral photoresponse is explored by gating graphene. We

find that doped graphene, under certain conditions, also acts effectively as a mirror by partially reflecting the polaritons. This modifies the hybridized modes, adding extra degrees of freedom in tuning the device photoresponse.

## Results

### Device configurations and their signal-to-noise ratios

We investigate five devices with their specifications and fabrication procedures described in Supplementary Note 1 and Methods, respectively. Initially, the optical response of these high-quality polaritonic nanoresonators is studied using FTIR, serving as a control experiment. For this purpose, we fabricate devices 1 and 4 that are used exclusively for FTIR measurements owing to their large area requirement (optical active area of ~ 30 × 30 μm², see optical image of device 1 in the inset of Fig. 1c and Supplementary Fig. 1 for device 4) and with a device configuration containing a single backgate to achieve uniform doping in graphene to maximize the optical response (see Supplementary Fig. 2, which indicates that the damping rate decreases with the increase in the Fermi level). However, this gating configuration does not allow the creation of a pn-junction for efficient photodetection[36]. Figure 1a (left panel) shows the schematic of devices 1 and 4, which consist of tens of nanometers wide metallic nanorods placed on top of hBN-encapsulated graphene. Upon illumination, scattering at the metallic rod array launches polaritons that propagate across the 2D

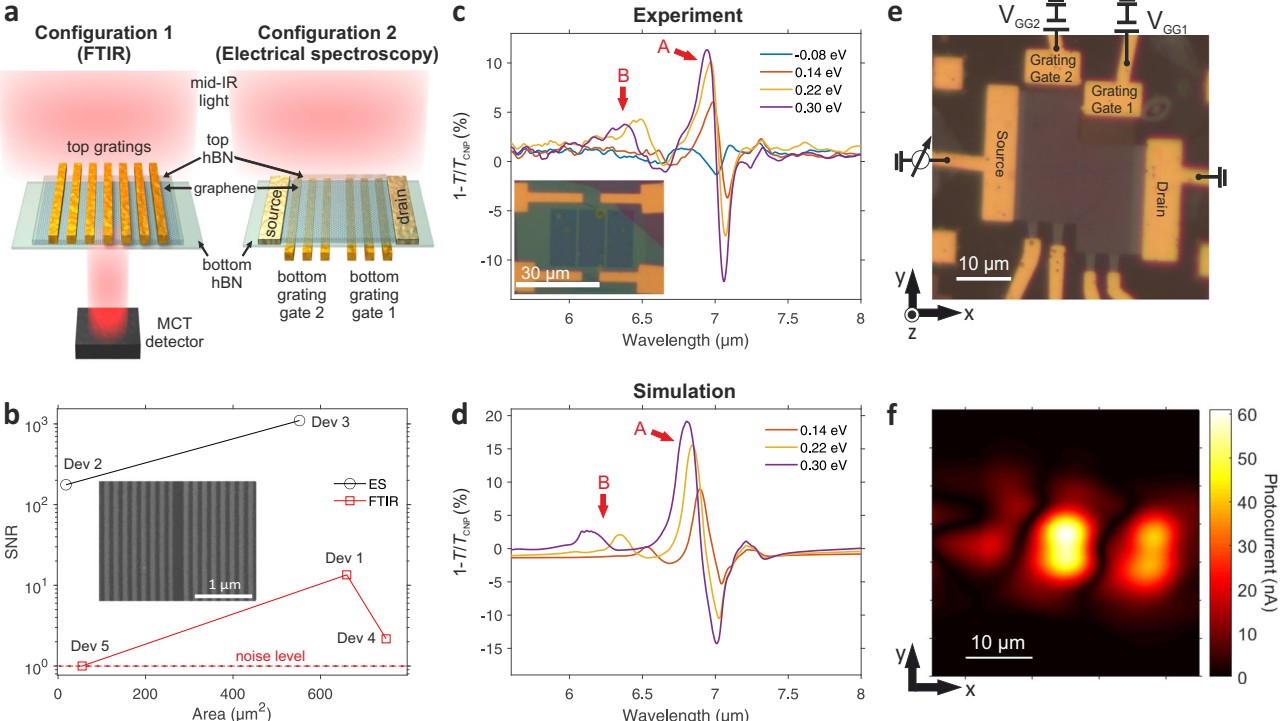

**Fig. 1 | Configurations of devices, transmission and photocurrent measurements, and optical simulation. a** Schematic representation of the measured devices (not to scale) consisting of two main configurations depending on the experiment. The 1st configuration (left panel) corresponds to that used exclusively for transmission measurements in Fourier Transform Infrared spectroscopy (FTIR), with the top metallic nanorods and the 2D stack below. A mercury-cadmium-telluride (MCT) detector is required to perform mid-infrared (mid-IR) spectroscopy. Devices 1 and 4 have this configuration. The 2nd configuration (right panel) consists of two grating bottom gates with a top 2D stack. Devices 2, 3, and 5 comprise this 2nd configuration. Devices 2 and 3 are measured using electrical spectroscopy, whereas device 5 is measured using FTIR. **b** The signal-to-noise ratio (SNR) of the five devices was measured using the corresponding technique, either by FTIR or photocurrent measurements (electrical spectroscopy). The noise level dashed line corresponds to an SNR of 1. The inset shows a Scanning electron

microscopy (SEM) image of the metallic nanorod array with a central gap for configuration 2, corresponding to devices 2, 3, and 5. **c** Extinction (1-$T/T_{CNP}$) spectrum of device 1 measured using FTIR, where $T$ and $T_{CNP}$ are the transmittances of the device at a certain gate voltage and at charge neutrality point (CNP), respectively. The curves correspond to several Fermi levels, as indicated in the legend. The inset shows the optical image of the device 1. The white scale bar corresponds to 30 μm. The three columns above the 2D stack are arrays of 100 nm wide metal nanorods with a 50 nm gap between them. A and B arrows indicate the polaritonic resonances described in the main text. **d** Finite-difference time-domain (FDTD) simulated extinction spectra of device 1 for several Fermi levels. **e** Optical image and device circuitry of configuration 2, which corresponds to device 3 used for photocurrent measurements. **f** Scanning photocurrent map (in absolute value) of device 3 at the incident wavelength ($\lambda$) of 6.6 μm. The gates are set to GG1 at 0.4 V and GG2 at − 0.25 V, thus creating a pn-junction.

heterostructure[6,30–32,34]. The graphene channel is uniformly doped by using a silicon backgate. To obtain a higher yield of fabrication of the metallic nanorods, we pattern them prior to the transfer of the 2D stack in devices 2 and 3, which are used exclusively for photocurrent measurements because they do not require a large optically active area (e.g., device 2 area of ~ 6 × 3 μm², see right panel of Fig. 1a and Supplementary Note 1 for more details). The metal nanostructures of these latter devices have a central gap (shown in the inset Fig. 1b) to split the array and gate the two graphene regions independently[36], thus creating a pn-junction. However, the grating gates produce a non-uniform electrostatic profile[44–46], which affects the damping rate and optical response, as shown in Supplementary Fig. 2. Device 3 is fabricated on an infrared transparent substrate (CaF₂) to investigate the hybridized polaritons in more detail since it avoids the presence of phonon polaritons near the hBN RBs spectral regions. We note that device 5 is based on configuration 2, but it is measured by FTIR, as shown in Supplementary Note 2.

The signal-to-noise ratio (SNR) values of the devices, measured using FTIR and electrical spectroscopy, are exhibited in Fig. 1b as a function of the device area. We notice that the devices measured by electrical spectroscopy show an SNR of 1 to 2 orders of magnitude higher than those measured by FTIR. For instance, device 5 shows an SNR of 1 because of its small area of 60 μm² for the FTIR requirements, whereas device 2 achieves an SNR of ~ 100 despite having the smallest area of 18 μm². These findings highlight the advantages of using electrical spectroscopy over standard FTIR for measuring polaritonic resonances in small active area devices. Further details of this comparison are provided in Supplementary Note 2.

## Optical spectroscopy in the mid-infrared range

Firstly, the optical response of the polaritonic nanoresonators is examined using FTIR to determine the extinction $1 − T/T_{CNP}$, where $T$ and $T_{CNP}$ are the transmittances of the device at a certain gate voltage and at charge neutrality point (CNP) respectively[6,25,32]. Figure 1c displays the extinction spectra for several Fermi energies. We identify two main peaks that exhibit a graphene plasmonic behavior, which increase their amplitude and blue shift (e.g., ≈ 0.15 μm for peak B) for increasing Fermi level[6]. Figure 1d depicts the simulated extinction using finite-difference time-domain (FDTD) as described in ref. 36 and semi-analytical rigorous coupled-wave analysis (RCWA) described in Supplementary Note 3. We observe excellent qualitative and quantitative agreement with the experimental results that we also support with the dispersion relation of the polaritonic modes present in device 1, as shown in Supplementary Fig. 3, Supplementary Note 3, and by showing the results of device 4 in Supplementary Fig. 5, therefore validating our theoretical model that will be explained in further detail below. The slightly lower experimental values are likely due to peak broadening caused by the inhomogeneity of metal rods' periodicity. This geometrical disorder impacts the scattering time of charge carriers in graphene, thus causing a decrease in peak intensity (see Supplementary Fig. 4). The absorption spatial profiles at the wavelengths of the measured peaks show a hybridized plasmon-phonon polariton for peak A, however, in peak B we do not observe a clear hybridization since the graphene plasmon mode resonates at its plane without interfering with the hBN HPPs (see Supplementary Fig. 6).

## Electrical detection of polaritonic nanoresonators

After determining the optical response and validating the theoretical model, we perform photocurrent spectroscopy measurements for electrical detection of the polaritonic nanoresonators. First, we perform a mid-infrared scanning photocurrent map across the device area shown in the optical image in Fig. 1e with its electrical circuitry. For this, we apply the appropriate voltages to dope the graphene region above the grating gate 1 (GG1) and 2 (GG2) with opposite polarity, creating a pn-junction in the graphene, for which the photocurrent is

maximum (see Fig. 1f and Supplementary Fig. 7). By tuning the gate voltages independently, we observe multiple sign changes of the photocurrent as shown in Supplementary Figs. 8, 9, which is consistent with the photothermoelectric (PTE) effect[15,16,36,47,48] with a responsivity of ~ 10 μA/W (see the spectral dependence of the responsivity and noise equivalent power in Supplementary Figs. 2 and 10 respectively). Further improvements in responsivity can be made through more optimized designs[36,49].

Next, by scanning the wavelength of the source, we spectrally resolve the resonances of the 2D polaritons from the normalized photocurrent spectra of device 2 (see Fig. 2a). The photocurrent spectra are normalized to the spectrum at CNP to probe the Fermi energy-dependent optoelectronic properties. GG1 is set to a fixed low voltage (doped region) since the Seebeck coefficient is maximum close to CNP[16,36], while GG2 is swept towards high negative voltages (doped p-type region), thus creating a doping asymmetry in the channel to maximize the photoresponse. We observe several peaks at the upper (≈ 6−7 μm) and lower (≈ 12−13 μm) RB of hBN and SiO₂ RB (≈ 8−9 μm). Some of these peaks at the RBs evolve with the Fermi level, which is ascribed to the hybridized plasmon-phonon polaritons. Moreover, two additional broader peaks (labeled as 4 and 8) appear at high Fermi energies outside these RBs. Their evolution and amplitude increase with the Fermi level more pronouncedly than those of the hybridized ones, which is in agreement with previous works[6,25].

To identify the origin of the resonances in the photocurrent spectrum, we show in Fig. 2b the simulated normalized absorption using FDTD and RCWA (see Supplementary Fig. 11). The absorption and photocurrent are proportionally related via the electronic temperature gradient[36], as shown in Supplementary Fig. 12. We find an excellent agreement in terms of spectral position and relative amplitude of the peaks. Additional peaks observed in the theoretical curves are due to resonances of higher-order modes[6]. We investigate the resonances in more detail by analyzing the field distributions above and outside the metal nanorods, as well as between the top and bottom layers of hBN, as shown in Fig. 2c−f. It can be seen that the field is confined to the bottom hBN above the metal, which corresponds to an acoustic graphene plasmon[6,18,32]. Conversely, between the metal nanorods, the field extends symmetrically in both hBN layers, as expected for conventional graphene plasmons[14,25]. Moreover, the electric field distribution (e.g., the $x −$ component of the field shown in Fig. 2g−j) indicates the resonance's harmonic order. For instance, peaks 4 and 8 show one sign change of the field in one period, as displayed in Fig. 2g, j (see also Supplementary Fig. 13), corresponding to the first harmonic. Different harmonic resonances appear at the top and bottom layers of hBN at peak 6 (Fig. 2i), which implies the superposition of the hybridized polaritonic modes. Another interesting case occurs at peak 5, where one period of the field matches two periods of the grating corresponding to a defect resonance (see Fig. 2h and Supplementary Fig. 14).

A complementary assessment involves calculating the dispersion relation of the polaritonic modes present in device 2 by using the transfer matrix method (TMM) described in Supplementary Note 3. The metallic rod arrays provide the in-plane effective momentum given by: $k_{eff}^l = k_{x,m}^l w/D + k_{x,g}^l g/D = 2\pi n/D$, where $D$, $w$, and $g$ are the periods, the width of the nanorod, and the gap between them, respectively. The parameter $l$ represents the order number of the mode, and $n \geq 1$ is the number of harmonic diffraction orders. The above equation encompasses the combination of both acoustic ($k_{x,m}$, above the metals) and conventional polaritons ($k_{x,g}$, above the gap)[31,34] as described in Supplementary Note 3 and Supplementary Fig. 15. The number of nodes present in the polaritonic field along the vertical direction determines $l$, while the number of nodes of the field along the lateral direction over a period determines $n$ (see also Supplementary Figs. 13 and 16). Noteworthily, $l = 0$ outside the RBs. We find a generally excellent correspondence with the experimental results as shown in

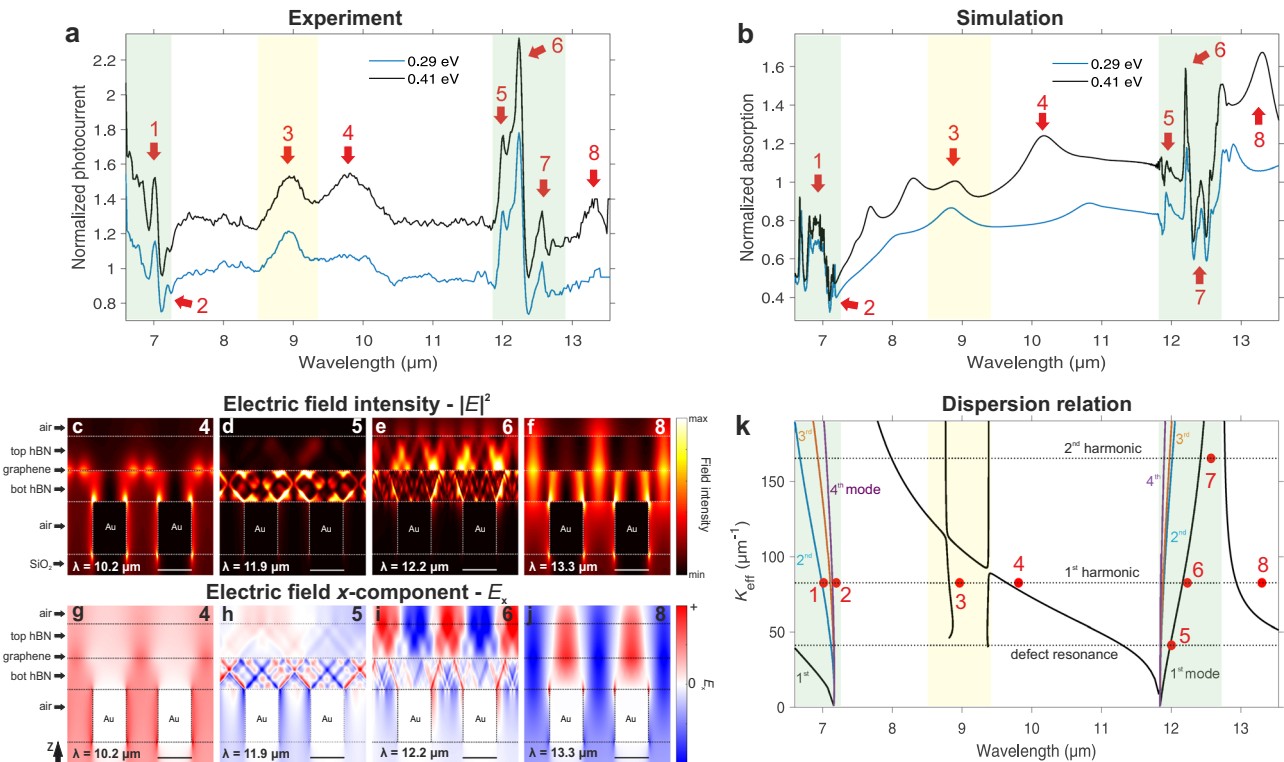

**Fig. 2 | Electrical spectroscopy measurements and simulations. a** Normalized photocurrent spectrum of device 2 at several Fermi energies. The photocurrent spectra are normalized to the spectrum at the charge neutrality point (CNP). The polaritonic peaks are labeled by red arrows. The highlighted spectral regions in green correspond to the upper and lower reststrahlen bands (RB) of hBN and, in yellow to the SiO₂ RB. The curves are offset for clarity. **b** Optical (FDTD) simulation of the graphene absorption spectrum for different Fermi energies normalized to the spectrum at CNP. We label the identified peaks in the same manner as the experimental ones in panel (**a**). **c–f** Cross-sectional view of the simulated electric field intensity normalized to the incident one across a region containing two metal nanorods, for wavelengths 10.2, 11.9, 12.2, and 13.3 μm corresponding to peaks 4, 5, 6, and 8, respectively in panel (**a**). The $x$ − (horizontal) and $z$ − (vertical) directions are defined in Fig. 1e. The white scale bar corresponds to 40 nm. The calculations

consider a non-uniform graphene Fermi level with a value of 0.4 eV above the metal (for a detailed doping profile, see Supplementary Fig. 2). **g–j** Same as panels (**c–f**), but the simulations instead show the cross-sectional view of the $x$-component of the electric field normalized to the incident one. The black scale bar corresponds to 40 nm. **k** Dispersion relation of the polaritonic modes with the respective harmonic diffraction orders ($2\pi n/D$). The three horizontal dashed lines correspond to the defect resonance ($n = 1/2$), first ($n = 1$), and second ($n = 2$) diffraction order resonances launched by the metal rod array, respectively. The marked red dots represent the experimental values, which the numeric labels are defined in Fig. 2a. The graphene Fermi level is 0.4 eV. At the hBN RBs (green highlighted regions), the black, blue, orange, and purple lines correspond to the 1ˢᵗ, 2ⁿᵈ, 3ʳᵈ, and 4ᵗʰ hybridized polaritonic modes, respectively. In yellow is highlighted the SiO₂ RB.

Fig. 2k. In the lower RB, for simplicity we consider the fundamental mode as the dominant, however, the polaritonic field distribution suggests a superposition of the fundamental with higher-order modes.

**Gate tunability of the hybridized polaritonic nanoresonators**

The hybridized polaritonic nanoresonators are investigated in more detail with device 3, which contains an infrared transparent substrate (CaF₂) to avoid phonon polaritons in the substrate near the hBN RBs spectral range, in particular at the lower RB (~ 12 μm)[50]. Following a similar procedure and gates' configuration explained previously, we tune the GG1 region to a fixed low n-type doping and sweep GG2 towards high p-type doping. At the upper RB (Fig. 3a), we identify several peaks of photocurrent whose amplitude increases and their spectral positions blue shift (e.g., 70 nm or 15 cm⁻¹ for peak 1′) with the increase of Fermi level. In fact, at the highest doping (0.35 eV), the value of normalized photocurrent at peaks 1′–3′ exceeds the values of the CNP curve, where the Seebeck coefficient is higher[36], thus showing the absorption enhancement caused by the polaritonic resonances. At the spectral location of the TO phonon of hBN, we observe a pronounced contribution at low Fermi level values which reduces its effect at high doping[51]. In the lower RB (Fig. 3b), we observe two main peaks that boost their amplitudes at higher Fermi levels, as well as a small blue shift

(e.g., 50 nm or 3 cm⁻¹ for peak 5′), corroborating the theoretical prediction shown in Supplementary Fig. 17.

The photocurrent peaks' locations agree well with the peaks in the simulated absorption spectra, shown in Fig. 3c, d. We point out a small redshift of ~ 0.15 μm for the spectral position of peaks at the lower RB, as well as a slight broadening of the experimental peaks compared to the theoretical ones. The latter is given by the inhomogeneity of the metallic rods' periodicity, as explained previously. Figure 3e, f shows that the maximum field occurs at the bottom hBN, which is between graphene and metallic nanorods, followed by partial transmission of HPPs rays towards the upper hBN. Hence, we have two different types of field distributions in each hBN layer, which are affected by the graphene doping. The dispersion relation of these hybrid polaritonic modes are shown in Supplementary Fig. 18 and demonstrates excellent agreement with experimental results.

To analyze the nature of this hBN-confined mode, which is controlled by the graphene Fermi energy, we compare two systems at the upper RB. The first one corresponds to the structure described earlier for device 2 (considering polaritons only above metal) shown in Fig. 4a, b, which include the spectral and spatial distributions of eigenmodes for the third and fifth modes, respectively. We notice that the third mode is characterized by two nodes in the vertical direction, while the fifth one has four nodes. In both cases, one of the nodes is located in

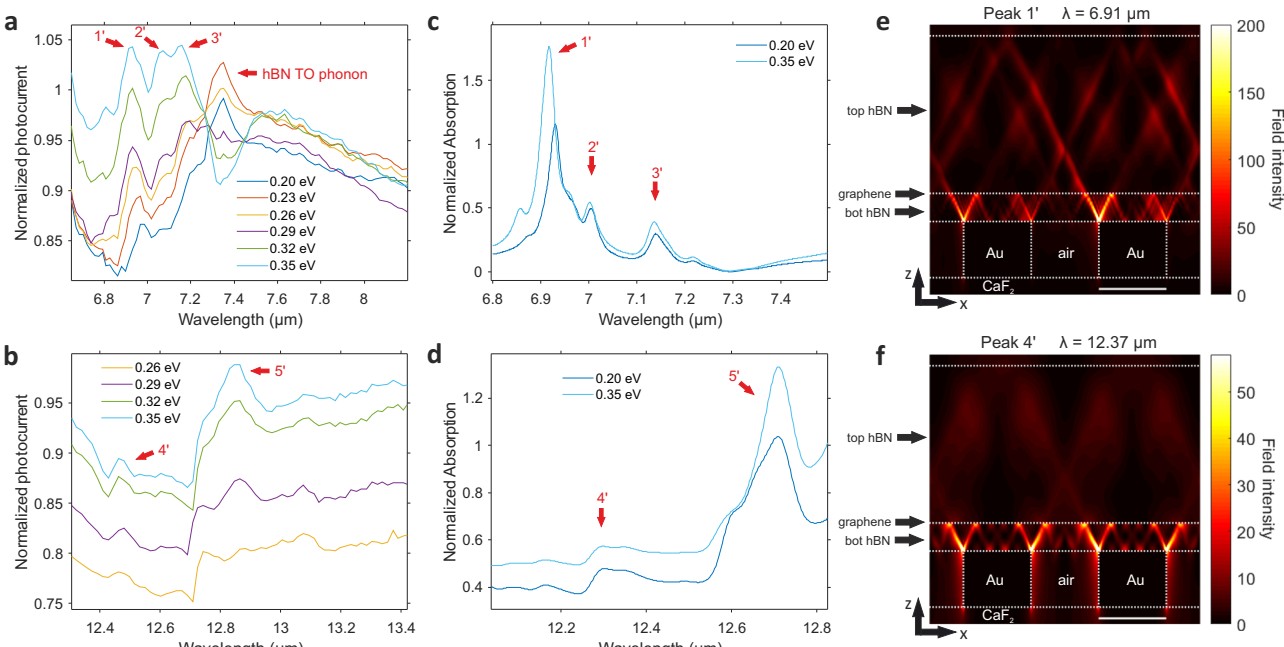

**Fig. 3 | Normalized photocurrent spectra of device 3 at the hBN RBs.**
**a** Normalized photocurrent spectra at the upper RB of hBN for several gate voltages. The photocurrent spectra are normalized to the spectrum at CNP. The polaritonic peaks are labeled by red arrows. The 0.20 eV curve is slightly shifted (divided by 1.05) for illustration. The Fermi energies are presented in absolute values. **b** Same as panel (**a**) but for the lower RB range. **c** Optical simulation of graphene absorption at the upper RB spectral region for different Fermi energies normalized to the spectrum at CNP. We label the identified peaks in the same manner as the experimental ones. **d** Same as panel (**c**) but for the lower RB range. **e** Cross-sectional view of the electric field intensity normalized to the incident one. The $x-$ (horizontal) and $z-$ (vertical) directions are defined in Fig. 1e. The simulations correspond to a non-uniform graphene Fermi level at 0.35 eV at wavelength 6.91 μm (corresponding to peak 1' in panel (**a**). The white scale bar corresponds to 20 nm. **f** Same as panel (**e**) but for a lower RB range at wavelength 12.37 μm (corresponding to peak 4' in Fig. 3b).

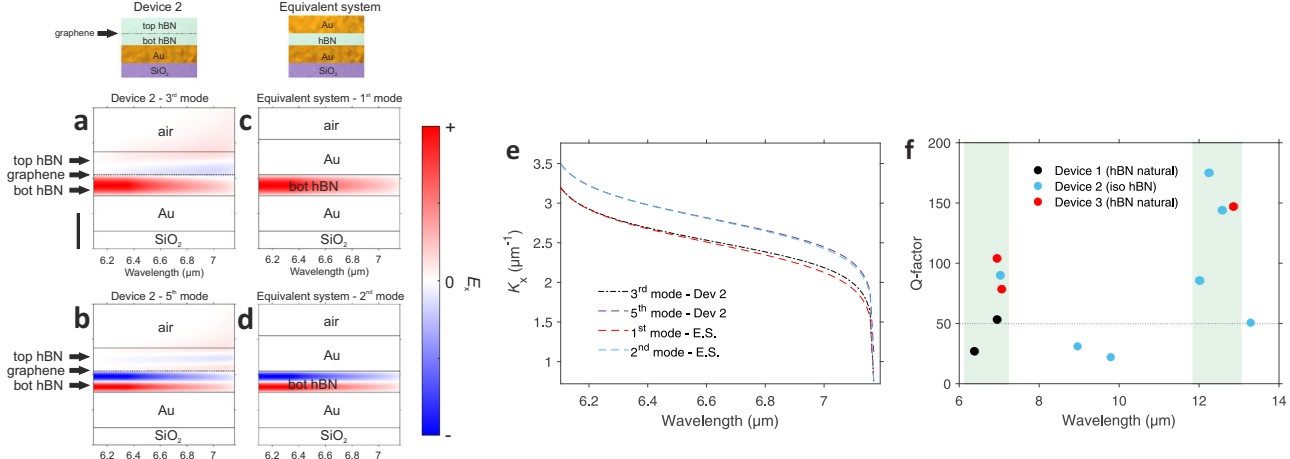

**Fig. 4 | Spatial field distribution and dispersion of hBN equivalent geometry and Q-factor spectrum of the 2D polaritonic nanoresonators.** Cross-sectional view of the spatial $x-$ component of the electric field as a function of the wavelength for (**a**) 3ʳᵈ and (**b**) 5ᵗʰ mode of the hybridized polariton. The graphene doping is 0.4 eV. The vertical black scale bar corresponds to 10 nm for panels (**a–d**). **c** and **d** correspond to the first and second order mode, respectively, of the bottom hBN
(6 nm thick) of the equivalent system (E.S.) geometry consisting of the bottom hBN embedded by two gold layers of 10 nm without the presence of graphene and the top hBN as shown in the illustrations on the top part of the figure. **e** Dispersion relation of the investigated modes of the two systems. **f** Q-factor spectrum of the measured 2D polaritonic nanoresonators. The regions highlighted in green correspond to the RBs of hBN.

the vicinity of graphene. In addition, the fields are mainly confined in the bottom hBN for both cases. The second system consists of replacing the graphene and top hBN layer with a gold film, as shown in Fig. 4c, d, along with their respective spatial distributions of fields from the two first eigenmodes. We observe similarities between the field distributions in the bottom hBN layer and those of the analogous geometry shown in Fig. 4a, b. This trend is further corroborated by comparing the dispersion curves of both systems, which overlap (see Fig. 4e). This model demonstrates that doped graphene acts as a mirror, partially reflecting polaritons in the bottom hBN layer.

Two of the key figures of merit for this resonant polaritonic system are the Q-factor and the mode volume or wavelength compression, as they are relevant for enhancing the light-matter interactions, for example, for sensing applications. However, the quality factors

diminish significantly when shrinking the dimensions of the nanoresonators to aim for deep subwavelength confinement[8,34]. The presented polaritonic platform overcomes this limitation by showing simultaneously resonances with Q-factor values up to ~ 200 inside the RBs (see Fig. 4f and Supplementary Fig. 19 for the Q-factor determination), due to the low loss nature of the hybridized polaritons[9,14] and an outstanding value of 330 is achieved for the optical lateral confinement or effective refractive index[34] ($n_{eff} \simeq k_p/k_{in}$, see Supplementary Fig. 20). In the case of graphene plasmons, we observe lower Q-factors up to 50, most likely limited by Ohmic losses, the non-uniform electrostatic potential and inhomogeneities in the metal nanorods' periodicity that act as additional scattering centers that reduce the resonant peak linewidth, as shown in Supplementary Fig. 2. In addition, we observe that Q-factor does not show significant variation as a function of Fermi level in agreement with other studies[25] (see Supplementary Fig. 21). We point out that the electrical spectroscopy approach enables the probing of very small nanoresonators (~ 30 nm), which is highly challenging for conventional techniques such as s-SNOM due to the resolution limitations imposed by the typical tip diameter (~ 50 nm)[8] or FTIR due to the sizeable area requirement (e.g., large arrays of nanoresonators)[6,31,32,34] as mentioned previously.

## Discussion

The investigated electro-polaritonic platform for performing mid and long-wave infrared photocurrent spectroscopy can be exploited to enhance photodetectors, hyperspectral and sub-diffraction imaging[52,53], and electrical detection of molecular vibrations and gases. The zero-bias operation of our device enables low noise and a significant reduction in power consumption. In addition, it operates at room temperature, in contrast to the MCT used in standard FTIR, which requires a voltage bias and liquid nitrogen cooling for optimum performance. We also highlight that our device is CMOS-compatible[54], which enables a highly compact platform that satisfies the size, weight, and power consumption (SWaP) requirements[55]. We further compare the electrical spectroscopy with standard FTIR in Supplementary Note 2.

These devices can also target other frequencies by tuning the size of the metallic nanorods and using different hyperbolic materials such as $\alpha$-MoO$_3$[56,57], V$_2$O$_3$[11], and black phosphorous. In particular, the in-plane anisotropy of MoO$_3$ could potentially change the interference and hybridization with graphene plasmons, thus modifying its effect on photodetection[58].

## Methods

### Device fabrication

The fabrication of devices 1 and 4, used for the transmission measurements, consists of the following: we first exfoliate the top and bottom hBN and the graphene onto freshly cleaned Si/SiO$_2$ substrates, stack them following the Van der Waals assembly technique[59,60] and transfer the hBN/graphene/hBN stack onto a high resistivity Si/SiO$_2$, which is a 50% transparent and gating capable substrate at the mid-infrared wavelengths[6,32]. We then use electron beam lithography (EBL) with a PMMA 950 K resist film to pattern source and drain electrodes and expose the device to a plasma of CHF$_3$/O$_2$ gases to partially etch the Van der Waals stack. Subsequently, we deposit side contacts of chromium (5 nm) / gold (60 nm) and lift off in acetone as described in ref. 59. Lastly, we pattern the nanorods with a period of 150 nm using EBL and deposit titanium (2 nm) / gold (8 nm) with a subsequent lift-off step in acetone.

Devices 2 and 3 were fabricated and designed for photocurrent measurements. On device 3, we first pattern the grating gates by using EBL and deposit titanium (2 nm) / gold (8 nm), followed by a lift-off step in acetone. Alternatively, for device 2, we pattern with Ga FIB (gallium-focused ion beam) a thin layer of gold deposited on a Si/SiO$_2$ substrate. The dimensions of the metallic nanorods are described in Supplementary Note 1. Following the previously mentioned procedure we transfer the hBN/graphene/hBN stack onto the grating gates. Then, we pattern the source and drain electrodes with a PMMA 950 K resist film using EBL and expose the patterned regions to a plasma of CHF$_3$/O$_2$ gases to partially etch the Van der Waals stack. Afterwards, we deposit side contacts of chromium (5 nm) / gold (80 nm) and lift off in acetone as described in ref. 59. Then, we define the hBN-encapsulated graphene channel by patterning a PMMA mask with EBL and etching it using a CHF$_3$/O$_2$ plasma. By performing electrical measurements using a 2-terminal configuration as a function of the gate voltages (varying GG1 and GG2 both at the same potential), we obtain 3000–15,000 cm²V⁻¹s⁻¹ as a lower bound of the estimated mobility (see Supplementary Fig. 2).

### Measurements

For the transmission measurements of devices 1, 4 and 5, we use a commercial FTIR (Fourier transform infrared) spectrometer (Bruker Tensor FTIR with a Bruker Hyperion 2000 microscope) and nitrogen-cooled mercury-cadmium-telluride (MCT) detector, which its spectral range goes from 6500 to 650 cm⁻¹ ($\lambda = 1.54$ to 15.4 μm) under normal incidence in air with p-polarized light (i.e., with incident polarization perpendicularly oriented respect to the main axis of the metallic gratings)[6]. We use a spectral resolution of 16 nm (4 cm⁻¹). We normalize the transmission spectrum with a reference signal at the graphene area with the backgate at the charge neutrality point (CNP) value (~ 0 V).

For the photocurrent spectroscopy measurements, we use a quantum cascade laser (QCL) mid and long-wave infrared laser (MIRcat from Daylight Solutions) with tunable wavelength ranges from 6.6 to 13.6 μm with a spectral resolution of < 1 cm⁻¹ and it's linearly polarized. We modulate the light via an optical chopper at 373 Hz, and we measure the photocurrent using a lock-in amplifier (Stanford Research). We scan the device position with a motorized xyz-stage. We focus the infrared light using a reflective objective with an NA of 0.5. To calibrate the incident power, we use a thermopile detector from Thorlabs placed at the sample location.

## Data availability

The data from this study is provided in the manuscript, Supplementary Information figures and from the corresponding authors upon request.

## Code availability

Codes from this work implementing the semi-analytical rigorous coupled-wave analysis (RCWA) and transfer matrix methods (TMM) (described in the Supplementary Information) are available upon request by contacting the corresponding authors.

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

## Acknowledgements

The authors thank Hanan Herzig Sheinfux, Krystian Nowakowski, and Iacopo Torre for fruitful discussions. F.H.L.K. acknowledges financial support from the Spanish Ministry of Economy and Competitiveness, through the "Severo Ochoa" Program for Centers of Excellence in R&D (SEV-2015-0522), support by Fundacio Cellex Barcelona, Generalitat de Catalunya through the CERCA program, and the Agency for Management of University and Research Grants (AGAUR) 2017 SGR 1656. Furthermore, the research leading to these results has received funding from the European Union Seventh Framework Program under grant agreements no.785219 and no. 881603 Graphene Flagship for Core2 and Core3. S.C. acknowledges financial support from the Barcelona Institute of Science and Technology (BIST), the Secretaria d'Universitats i Recerca del Departament d'Empresa i Coneixement de la Generalitat de Catalunya and the European Social Fund (L'FSE inverteix en el teu futur) - FEDER. N.M.R.P. acknowledges support from the Independent Research Fund Denmark (grant no. 2032-00045B) and the Danish National Research Foundation (Project No. DNRF165). Y.V.B., M.I.V., and N.M.R.P. acknowledge support by the Portuguese Foundation for Science and Technology (FCT) in the framework of the Strategic Funding UIDB/04650/2020. K.W. and T.T. acknowledge support from the JSPS KAKENHI (Grant Numbers 21H05233 and 23H02052) and World Premier International Research Center Initiative (WPI), MEXT, Japan, for the growth of h-BN crystals. Funding for hBN crystal growth by E.J. and J.H.E. was provided by the Office of Naval Research, Award no. N00014-22-1-2582. F.H.L.K. and S.C. acknowledge funding from the European Union (ERC, POLARSENSE, 101123421). Views and opinions expressed are, however, those of the author(s) only and do not necessarily reflect those of the European Union or the European Research Council Executive Agency. Neither the European Union nor the granting authority can be held responsible for them.

## Author contributions

S.C., D.A.I., and F.H.L.K. conceived the project. S.C. and H.A. fabricated the devices and performed the experiments. D.A.I. and A.G. assisted in the experiments. M.C. and R.K.K. supported the device fabrication. I.V., Y.V.B., N.M.R.P., and E.L. performed the simulations and developed the theoretical model. S.C. and M.I.V. assisted in the modeling. S.C., I.V., Y.V.B., E.L., and F.H.L.K. wrote the manuscript. K.W. and T.T. synthesized the hBN crystals. E.J. and J.H.E. synthesized the isotopically enriched hBN crystals. N.M.R.P., E.L., and F.H.L.K. supervised the work and discussed the results. All authors contributed to the scientific discussion and manuscript revisions. S.C., H.A., I.V., and Y.V.B. contributed equally to the work.

## Competing interests

The authors declare no competing interests.
