## [Peer Review File · Nature Communications]

REVIEWER COMMENTS

Reviewer #2 (Remarks to the Author):

I thank the authors for the detailed response to my comments. I support publication of the manuscript as is.

Reviewer #4 (Remarks to the Author):

In this manuscript, the authors realized the electrical spectroscopy of deep sub-wavelength polaritonic nanoresonators in a compact platform. The configuration efficiently converts infrared light into an electrical signal and is promising for integrated photo-detection and sensing applications since no external detector is required. I have no doubt about the novelty of this work. Once the authors are able to address the following comments, I can recommend its publication in Nature Communications.

1. Due to the multiple devices and complicated structure of the manuscript, it is a bit difficult for the readers to follow its logic. For example, in Fig.1, the authors show a schematic of device-1 in Fig.1a, an SEM picture of device-2 in Fig.1b, an optical image of device-3 in Fig.1c, and an optical image of device-1 in the inset of Fig.1d. I strongly suggest that the author rearrange the figures in the manuscript and make it easier for the readers. One suggestion: since one of the main novelties of this work is that electrical measurements require only a small area, the authors might compare the configurations of the different devices in Fig.1, rather than just show them in the Supplementary Information.

2. The authors claim that the electrical spectroscopy technique holds great promise for a variety of applications. However, they do not give any performance evaluation criteria, e.g., signal to the noise etc. They should compare the key parameters of the photocurrent spectroscopy techniques and traditional techniques, e.g., FTIR.

3. Why the authors show different spectral ranges for FTIR and photocurrent spectroscopy in the manuscript? For example, the authors show FTIR results in the range from 6 to 8 μm in Fig.1, photocurrent results in the range from 6.5 to 13.5 μm in Fig.2, photocurrent results in the range from 6.8 to 8.5 μm and 12.4 to 13.4 μm in Fig.3.

4. How did the authors calculate the quality factors?

5. I agree with the reviewer 2 that the authors should clearly compare the FTIR results with the photocurrent spectra. I am not convinced by the authors' response that different measurements require different samples. Is it possible to prepare a sample with large areas, one part of which is used for electrical measurements and the other for FTIR measurements (The two parts can be disconnected to avoid interfering with each other)?

Or is it possible to prepare two samples with similar structures, e.g., similar h-BN thicknesses?

REVIEWER COMMENTS

Reviewer #2 (Remarks to the Author):

I thank the authors for the detailed response to my comments. I support publication of the manuscript as is.

We thank the reviewer for his/her valuable comments and inputs, which have significantly helped us improve the manuscript.

Reviewer #4 (Remarks to the Author):

In this manuscript, the authors realized the electrical spectroscopy of deep sub-wavelength polaritonic nanoresonators in a compact platform. The configuration efficiently converts infrared light into an electrical signal and is promising for integrated photo-detection and sensing applications since no external detector is required. I have no doubt about the novelty of this work. Once the authors are able to address the following comments, I can recommend its publication in Nature Communications.

We thank the reviewer for highlighting the novelty of our study.

1. Due to the multiple devices and complicated structure of the manuscript, it is a bit difficult for the readers to follow its logic. For example, in Fig.1, the authors show a schematic of device-1 in Fig.1a, an SEM picture of device-2 in Fig.1b, an optical image of device-3 in Fig.1c, and an optical image of device-1 in the inset of Fig.1d. I strongly suggest that the author rearrange the figures in the manuscript and make it easier for the readers. One suggestion: since one of the main novelties of this work is that electrical measurements require only a small area, the authors might compare the configurations of the different devices in Fig.1, rather than just show them in the Supplementary Information.

We appreciate the reviewer's comments. We have improved the clarity of the configurations of the devices by changing the Figure 1a with the following one shown below for explaining the different configurations used for the experiments performed.

We also rearranged the panels in Figure 1 to enhance readability so that it chronologically explains the different device configurations with their respective measurements.

We have also followed the suggestion of the reviewer and added panel b, which compares the different devices by exhibiting their SNR as a function of the device area for both FTIR and electrical spectroscopy.

Figure 1. Configurations of devices, transmission and photocurrent measurements, and optical simulation. a) Schematic representation of the measured devices (not to scale) consisting of two main configurations depending on the experiment. The 1st configuration (left panel) corresponds to that used exclusively for transmission measurements in FTIR, with the top metallic nanorods and the 2D stack below. An MCT detector is required to perform mid-IR spectroscopy. Devices 1 and 4 have this configuration. The 2nd configuration (right panel) consists of two grating bottom gates with a top 2D stack. Devices 2, 3, and 5 comprise this 2nd configuration. Devices 2 and 3 are measured using electrical spectroscopy, whereas device 5 is measured using FTIR. b) The signal-to-noise ratio (SNR) of the five devices measured using the corresponding technique, either by FTIR or photocurrent measurements (electrical spectroscopy). The noise level dashed-line corresponds to an SNR of 1. The inset shows an SEM image of the metallic nanorod array with a central gap for configuration 2, corresponding to devices 2, 3 and 5. c) Extinction ($1-T/T_{CNP}$) spectrum of the device 1 measured using FTIR. The curves correspond to several Fermi levels as indicated in the legend. The inset shows the optical image of the device 1. The white scale bar corresponds to $30\ \mu\text{m}$. The three columns above the 2D stack are arrays of $100\ \text{nm}$ wide metal nanorods with a $50\ \text{nm}$ gap between them. d) FDTD simulated extinction spectra of device 1 for several Fermi levels. e) Optical image and device circuitry of configuration 2, which corresponds to device 3 used for photocurrent measurements. f) Scanning photocurrent map (in absolute value) of device 3 at the incident wavelength (λ) of $6.6\ \mu\text{m}$. The gates are set to GG1 at $0.4\ \text{V}$ and GG2 at $-0.25\ \text{V}$, thus creating a pn-junction.

2. The authors claim that the electrical spectroscopy technique holds great promise for a variety of applications. However, they do not give any performance evaluation criteria, e.g., signal to the noise etc. They should compare the key parameters of the photocurrent spectroscopy techniques and traditional techniques, e.g., FTIR.

In the revised version of the manuscript and supplementary information, we have compared the different devices that we measured to illustrate the signal-to-noise ratio (SNR) as a function of various device parameters, such as device area, resistance, and grating period, etc. We also evaluated the noise in the FTIR and photocurrent measurements for different characteristics of the devices. To support this further, we have added the results of a device labeled as device 5, which has similar characteristics to device 2, but the SNR and noise measured in the FTIR are extremely small and high,

respectively. We have added a section in the revised version of the SI, labeled Supplementary Note 2, describing all of these in detail.

In addition to this comparison, we stress that in the FTIR measurements shown in Fig. 1c and Supplementary Figure 5a, we have performed a smoothing procedure for the curves by using a mean moving filter approach despite the averaging step of repetitive scans (usually we take 5 to 50 at the same gate voltage), whereas in the photocurrent measurements, we use the raw data. In Supplementary Note 2, we show the plots of FTIR without the smoothing process and mention the corresponding SNR. This is discussed in more detail in that section.

We highlight that it is difficult to observe polaritonic resonances in the lower RB spectral range in FTIR spectroscopy because of the high noise level in all the measured devices. This could be due to the low detectivity in this spectral range of the MCT detector, which is typically used in commercial FTIR. Therefore, owing to the broadband absorption spectrum of graphene, this is not an issue in the electrical spectroscopy technique combined with the high SNR, as shown in Supplementary Figures 28 and 29. Thus, in the electrical spectroscopy measurements, we probe the polaritonic resonances in the lower RB, as shown in the main text for devices 2 and 3.

We have added the following sentences in the main text in the discussion section to remark the advantages of the electrical spectroscopy technique with respect to the standard FTIR:

"The zero-bias operation of our device enables low noise and a significant reduction in power consumption. In addition, it operates at room temperature, in contrast to the MCT used in standard FTIR, which requires a voltage bias and liquid nitrogen cooling for optimum operation. We also highlight that our device is CMOS-compatible, which enables a highly compact platform that satisfies the size, weight, and power consumption (SWaP) requirements. We further compare the electrical spectroscopy with standard FTIR in the Supplementary Note 2."

We have added a table in the Supplementary Note 2 that compare the detector part of the electrical spectroscopy approach with traditional FTIR systems:

Parameter	FTIR spectroscopy	Electrical spectroscopy
Voltage bias required	Yes	No
Cooling of the detector	Yes, with liquid nitrogen	No, room temperature operation
Smoothing of the data	Yes	No
Maximum SNR measured	13	1100
Minimum measurable sample area	660 μm^2	18 μm^2
CMOS compatibility	No	Yes
Size	Bulky	Compact
Fulfillment of SWaP	No	Yes
Operating wavelength	up to $\sim 12 \mu\text{m}$	No wavelength cut-off

Supplementary Table II. FTIR and electrical spectroscopy comparison of the main detector parameters.

3. Why the authors show different spectral ranges for FTIR and photocurrent spectroscopy in the manuscript? For example, the authors show FTIR results in the range from 6 to 8 μm in Fig.1, photocurrent results in the range from 6.5 to 13.5 μm in Fig.2, photocurrent results in the range from 6.8 to 8.5 μm and 12.4 to 13.4 μm in Fig. 3.

In the case of the FTIR measurements of devices 1 and 4 shown in Figure 1c-d and Supplementary Figure 5, respectively, we focus on the small spectral range of 6-8 μm because of the high noise outside that range, as shown in Supplementary Note 2. Therefore, for clarity, we mainly show the spectral range in which the polaritonic resonances are clearly observed. In Supplementary Note 2, we show the full range of the extinction spectrum for devices 1, 4, and 5 measured by FTIR, so that the SNR and noise with this technique are exhibited.

In addition, we point out that in the FTIR measurements, the MCT detector limits the measurable spectral range, which becomes quite noisy above 12 μm , as shown in Supplementary Note 2. However, our graphene detector used in the electrical spectroscopy measurements has broadband operation and low noise; thus, it does not present this spectral limitation.

Regarding the figures of the photocurrent measurements performed on devices 2 and 3, the spectral range was limited by the laser, which ranges between 6.6 and 13.6 μm as described in the Methods section. Therefore, we cannot present the same range as that measured using FTIR.

In the case of the photocurrent measurements in device 3 shown in Figure 3, we focused on the lower and upper RBs of the hBN spectral ranges to show the gate tunability of these hybridized polaritonic resonances. In addition, a CaF_2 substrate was used to avoid optical phonon losses of SiO_2 above 8 μm , as explained in the main text, and to obtain a high SNR in the lower RB spectral range, as shown in Supplementary Fig. 28, which has been added to Supplementary Note 2. To show the gate tunability, we took a smaller spectral range and focused on the upper and lower RB of hBN to efficiently acquire high-resolution spectra while changing the gate voltage to several values. Using this approach, we avoid any potential gate hysteresis effects. We discarded any polaritonic resonance between RBs using preliminary measurements.

We have added plots containing the entire measurable spectral range of devices 1 to 5 in Supplementary Note 2 added in this revised version of the SI, and we discuss further details about the SNR of each technique.

4. How did the authors calculate the quality factors?

We fitted the polaritonic peaks with a Lorentzian function to determine their full-width-half-maximum (FWHM). We divided the wavelength or wavenumber value of the central position of the peak by the FWHM obtained. We have added a reference to the SI of this description in the main text of the manuscript and an example to Supplementary Figure 19 in the SI, as shown below.

Example:

Supplementary Figure 19. Fit of polaritonic peaks of the normalized photocurrent. The fit corresponds to peaks 3 and 4 of device 2 shown in Figure 2a of the main text. We use a Lorentzian fit (black line) and we obtain a FWHM of 35.9 and 46.8 at 1115.6 cm^{-1} (8.96 μm) and 1021.1 cm^{-1} (9.79 μm) for peak 3 and 4, respectively (data shown in red open circles). By dividing the peak's wavenumber (wavelength) central position value by its FWHM, we obtain Q factors of 31 and 22 for peaks 3 and 4, respectively.

5. I agree with the reviewer 2 that the authors should clearly compare the FTIR results with the photocurrent spectra. I am not convinced by the authors' response that different measurements require different samples. Is it possible to prepare a sample with large areas, one part of which is used for electrical measurements and the other for FTIR measurements (The two parts can be disconnected to avoid interfering with each other)?

Or is it possible to prepare two samples with similar structures, e.g., similar h-BN thicknesses?

The primary purpose of this study is not to provide a direct comparison between FTIR and photocurrent measurements. Instead, FTIR measurements present preliminary results that probe polaritonic resonances in high-mobility hBN-encapsulated graphene devices, which have not been previously reported in the literature in the far-field mid-infrared range with hybridized polaritons. These transmission measurements focus solely on the optical response of the device and do not account for the contributions from the photocurrent. The optical results validate the complex theoretical model that incorporates acoustic and conventional graphene plasmons and/or acoustic and conventional hybridized polaritons, a non-uniform Fermi level across the graphene channel, and absorption broadening due to the size distribution of the nanorods. This model enabled us to describe the photocurrent measurements, as demonstrated in Supplementary Figure 12, where the absorption allows the determination of the electronic temperature in graphene and, consequently, the responsivity.

In response to the referee's first suggestion regarding the use of samples with larger areas, the largest fabricated and measured device (device 4), with an area exceeding 750 μm^2 (30 \times 25 μm^2), exhibited an FTIR signal that was obscured by noise. The signal only becomes distinguishable after smoothing the curves, resulting in a very small extinction value of $\sim 2\%$, as shown in the optical results in Supplementary Figures 5 and 22 and Supplementary Note 2. Therefore, even though we successfully fabricated a large-area device, which is highly challenging, as explained below, it is not guaranteed that we would be able to resolve the same features observed with the electrical spectroscopy technique. Additionally, as previously mentioned, it is very difficult to observe clear features of polaritonic resonances in the lower RB spectral range in FTIR because of the generally reduced performance of MCT detector in this wavelength range.

A potential device capable of yielding accurate results in FTIR spectroscopy would require an optically active area of approximately 100 \times 100 μm^2 , which would allow the use of an aperture of this size to

achieve a good SNR in FTIR measurements^[1], as demonstrated by Iranzo, D.A. et al. Science 2018. However, these large areas can be achieved with scalable CVD graphene, as demonstrated by Iranzo, D.A. et al. Science 2018. While such large areas are attainable using scalable CVD graphene, as shown by Iranzo, D.A. et al., achieving scalable CVD hBN growth with thicker, flat, and more uniform layers than a monolayer remains a significant challenge (see references Shen, Y. et al., Advanced Materials 2021; Knobloch, T. et al., Nature Electronics 2021). Additionally, the use of CVD-based materials results in low Q factors for polaritonic nanoresonators, with values below 10, and low mobility in graphene, as reported by Iranzo, D.A. et al., Science 2018 and Epstein, I. et al., Science 2020.

In our work, we demonstrate high-quality polaritonic resonances by using pristine materials to efficiently exploit their optoelectronic properties not only for optical spectroscopy but also for photodetection. The photothermoelectric (PTE) effect, which enables the electrical detection of polaritons, is stronger in high-mobility graphene (see references Castilla, S. et al. Nature Comm 2020; Asgari, M. et al. Applied Physics Letters 2022). We anticipate that future advancements in this technology will enable the development of scalable, high-quality CVD hBN, and high-mobility graphene-based devices.

Fabricating an hBN-encapsulated exfoliated graphene stack with a large area is highly challenging because obtaining homogeneous flakes of this size is a time-consuming process. Moreover, the stacking procedure of these large and thin flakes consists of a low-yield process, requiring careful handling to avoid cracks in the flakes and to ensure a bubble-free area of those dimensions (see references Pizzocchero, F. et al. Nature Comm 2016 ; Purdie, D.G., et al. Nature Comm 2018). To the best of our knowledge, a device based on an exfoliated hBN-encapsulated graphene structure with the previously mentioned requirements and an area of $100 \times 100 \mu\text{m}^2$ has not been reported in the literature, highlighting the extreme difficulty of this fabrication process.

Besides the previous point, the effective fabrication of top metallic nanorods to achieve a uniform Fermi level in graphene using a silicon or graphite back gate to enhance the optical response (as shown in Supplementary Figures 2 and 15) would require a specific dose for patterning them that depends on the hBN thickness. This would result in an even lower fabrication yield, as we encountered during the fabrication process of non-successful devices that are not reported in this work. Supplementary Figure 1 shows the images of devices 1 and 4, highlighting the lift-off issues with these top metallic nanorods. A sonication step was required, which could potentially damage the quality of the 2D stack. Owing to these lift-off problems, we decided to switch to bottom gratings to obtain a higher fabrication yield.

For photocurrent operation, these bottom gratings require a gap at the center of the array to connect them separately to two electrodes, allowing the application of opposite voltages to create a graphene pn-junction. As explained in the main text, these bottom gratings result in decreased absorption across the device compared with the configuration with top metallic nanorods and a uniform Fermi level in graphene, as shown in Supplementary Figures 2 and 15. Although the decrease in absorption in graphene is a significant issue for potential FTIR measurements, it does not represent a problem for electrical spectroscopy measurements, because the SNR is very high.

In addition, using a hypothetical device with the same configuration as that used for photocurrent measurements but designed to cover a large area (e.g., $100 \times 100 \mu\text{m}^2$), as in the optimized FTIR setup mentioned previously, would result in electrodes positioned at the edges and separated by $100 \mu\text{m}$. This design would produce a highly resistive device, leading to an inefficient configuration to exploit the PTE effect (see ref. Castilla, S. et al. Nature Comm 2020). As a result, the SNR and responsivity in the electrical spectroscopy measurements would decrease, leading to ineffective electrical detection of the polaritonic resonances. One potential solution is to design arrays of photodetectors that cover

a large area. However, implementing this approach would require additional design and simulation steps that are beyond the scope of this work.

Following the 2nd suggestion made by the reviewer, we have added the results of FTIR measurements of a 5th device that has very similar characteristics to device 2 (measured by electrical spectroscopy) in terms of the device area, grating period, hBN thicknesses, etc., as shown in Supplementary Figure 24 and Supplementary Note 2. This 5th device has the same configuration 2 with the bottom metallic grating gates of devices 2 and 3, and the latter two devices were designed and fabricated for photocurrent measurements. Although the resistance of graphene is effectively tunable by the gates, we were not able to identify any polaritonic resonance because of the high noise and low SNR of this device owing to its small active area, as shown in Supplementary Note 2 added to this revised version of the SI. We note that we could not perform electrical spectroscopy measurements afterwards because the device stopped working during the FTIR measurements as we applied high voltages in the gates to probe high Fermi energies. On the other hand, by performing optical simulations in RCWA, we show that polaritonic resonances would be expected in the measured wavelength range and also how these resonances are hidden by introducing noise in the simulated extinction spectrum. These simulations have been added in Supplementary Figures 25 and 26. These results highlight the SNR limitations of FTIR for small devices.

Additional References

[1] ThermoFisher Scientific. FTIR Microscopy: How experimental decisions affect the signal-to-noise ratio. <http://www.thermofisher.com/RaptIR>.

REVIEWERS' COMMENTS

Reviewer #4 (Remarks to the Author):

The authors successfully addressed all my concerns and I support its publication in Nature Communicaitons